# Root Zone Water Management Effects on Soil Hydrothermal Properties and Sweet Potato Yield

**DOI:** 10.3390/plants13111561

**Published:** 2024-06-05

**Authors:** Shihao Huang, Lei Zhao, Tingge Zhang, Minghui Qin, Tao Yin, Qing Liu, Huan Li

**Affiliations:** College of Resources and Environmental Sciences, Qingdao Agricultural University, Qingdao 266109, China; hsh25495@163.com (S.H.); zl199511231111@163.com (L.Z.); zhangtingge0929@163.com (T.Z.); 19861607932@163.com (M.Q.); yintao@qau.edu.cn (T.Y.); qy7271@163.com (Q.L.)

**Keywords:** sweet potato, soil volumetric water content, soil temperature, soil CO_2_ concentration, root morphology, physiological characteristics

## Abstract

Sufficient soil moisture is required to ensure the successful transplantation of sweet potato seedlings. Thus, reasonable water management is essential for achieving high quality and yield in sweet potato production. We conducted field experiments in northern China, planted on 18 May and harvested on 18 October 2021, at the Nancun Experimental Base of Qingdao Agricultural University. Three water management treatments were tested for sweet potato seedlings after transplanting: hole irrigation (W_1_), optimized drip irrigation (W_2_), and traditional drip irrigation (W_3_). The variation characteristics of soil volumetric water content, soil temperature, and soil CO_2_ concentration in the root zone were monitored in situ for 0–50 days. The agronomy, root morphology, photosynthetic parameters, ^13^C accumulation, yield, and yield components of sweet potato were determined. The results showed that soil VWC was maintained at 22–25% and 27–32% in the hole irrigation and combined drip irrigation treatments, respectively, from 0 to 30 days after transplanting. However, there was no significant difference between the traditional (W_3_) and optimized (W_2_) drip irrigation systems. From 30 to 50 days after transplanting, the VWC decreased significantly in all treatments, with significant differences among all treatments. Soil CO_2_ concentrations were positively correlated with VWC from 0 to 30 days after transplanting but gradually increased from 30 to 50 days, with significant differences among treatments. Soil temperature varied with fluctuations in air temperature, with no significant differences among treatments. Sweet potato survival rates were significantly lower in the hole irrigation treatments than in the drip irrigation treatments, with no significant difference between W_2_ and W_3_. The aboveground biomass, photosynthetic parameters, and leaf area index were significantly higher under drip irrigation than under hole irrigation, and values were higher in W_3_ than in W_2_. However, the total root length, root volume, and ^13^C partitioning rate were higher in W_2_ than in W_3_. These findings suggest that excessive drip irrigation can lead to an imbalance in sweet potato reservoir sources. Compared with W_1_, the W_2_ and W_3_ treatments exhibited significant yield increases of 42.98% and 36.49%, respectively. The W_2_ treatment had the lowest sweet potato deformity rate.

## 1. Introduction

Despite the continuous development of agricultural irrigation systems, the irrational use of water resources has led to imbalances in water supply and demand that have gradually become a key factor restricting agricultural production in China [1]. The efficient utilization of water resources has become a crucial issue for sustainable agricultural development and improvement of the ecological environment [2].

Sweet potato is a drought-resistant crop that tolerates both waterlogging and drought. Natural precipitation is a crucial water source for sweet potato, and excessive or insufficient water supply can have adverse effects on its growth [3]. Northern China is among the main sweet potato production areas in China, with harvests typically conducted in the spring and summer. Water supply is the main factor limiting sweet potato production in northern China [4]. Watering seedlings after transplanting is crucial for their rooting and branching, and sweet potato fields in northern China are generally arid or semi-arid in spring, resulting in water shortages due to insufficient precipitation. In the past, sweet potato cultivation in China has often employed hole irrigation for transplanted seedlings; however, this method is time-consuming, laborious, and susceptible to drought stress, resulting in poor and unstable yields. In recent years, there has been an attempt to introduce drip irrigation for sweet potato seedling transplanting. Drip irrigation is preferred because it does not damage the soil structure and helps to maintain soil water, gases, and heat at suitable conditions. Additionally, this approach conserves both labor and water [5]. However, sweet potato producers tend to believe that more drip irrigation promotes seedling survival, resulting in excessive irrigation due to a lack of appropriate scientific guidance on seedling water amounts.

Soil is a complex medium composed of solid, liquid, and gaseous phases, which must be maintained in appropriate proportions to form a favorable environment for crop growth [6]. Studies have shown that excessive single-application irrigation in relatively permeable soil causes water to leak into deeper layers, keeping irrigation water away from areas where roots are concentrated and hindering the efficient use of irrigation water [7,8,9]. In addition, when soil water levels are too high, poor aeration can inhibit root respiration and slow the belowground development of sweet potatoes. In severe cases, over-irrigation can cause root rot, which reduces seedling growth rates [10,11,12]. Conversely, insufficient irrigation to meet the water demand of sweet potato seedlings may increase soil oxygen content, improving soil aeration but resulting in stunted crop roots [13,14].

A previous study altered soil water content, aeration, and soil temperature through the application of different irrigation levels after sweet potato transplantation; changes to the soil environment of the root zone indirectly affected the above- and belowground growth of sweet potato seedlings [15]. Other studies have primarily conducted pot experiments to examine individual soil water, air, or heat effects, without examining interactions among multiple factors. Additionally, few researchers have explored the spatiotemporal variability of sweet potato responses to factors related to soil water, air, and heat, or the impact of the root zone soil environment on sweet potato growth in domestic and international contexts. Therefore, in the present study, we conducted field experiments and in situ monitoring of various factors to investigate the impacts of soil water, aeration, and heat on sweet potato seedling growth to reveal their intrinsic relationships with sweet potato physiology. We predicted that water had a direct effect on the growth and yield of sweet potato, while temperature had an indirect effect. Our findings provide a theoretical basis by which to guide the rational irrigation of sweet potato plants after transplantation.

## 2. Results

### 2.1. Soil Volumetric Water Content (VWC), Soil Temperature, and Soil CO_2_ Content Dynamics

Soil VWC increased as the amount of water provided to transplanted seedlings increased; this increase was significantly higher under drip irrigation than under hole irrigation (*p* < 0.05). However, there was no significant difference between optimized and traditional drip irrigation methods (*p* > 0.05). VWC remained stable in all treatments for the first month after transplantation; from 30 to 50 days, VWC gradually decreased during the fertility period, with significant differences among treatments (*p* < 0.05).

The soil temperature in the 0–20 cm soil layer increased over time, but with non-significant differences among the three treatments (*p* > 0.05). Significant cooling occurred on days 21 and 28, but the temperature quickly recovered thereafter. By day 35, the treatments had stabilized. These results suggest that soil temperature was primarily influenced by changes in atmospheric temperature.

Soil CO_2_ content decreased as the irrigation amount increased. In the first 30 days, drip irrigation resulted in a significantly higher CO_2_ content than hole irrigation (*p* < 0.05), whereas no significant difference was detected between optimized and conventional irrigation methods (*p* > 0.05). From 30 to 50 days after transplantation, soil CO_2_ content was influenced by soil moisture content and plant root respiration, decreasing in the order W_2_ > W_3_ > W_1_ (Figure 1).

### 2.2. Treatment Effects on Seedling Survival Rates, Biomass, Leaf Area Index (LAI), and Photosynthetic Parameters

#### 2.2.1. Survival Rates

The survival rates of treatments W_1_, W_2_, and W_3_ were 93.6%, 96.48%, and 96.64%, respectively. Survival rates increased with the irrigation amount. Drip irrigation led to significantly higher survival rates than hole irrigation (*p* < 0.05). However, there was no significant difference between optimized and conventional irrigation methods (*p* > 0.05). These results demonstrate that drip irrigation was more effective than hole irrigation but that drip irrigation did not significantly increase the seedling survival rate due to over-irrigation (Figure 2).

#### 2.2.2. Biomass

The above- and belowground dry weight biomass of seedlings increased as the irrigation amount increased in all three sampling periods. Both the above- and belowground dry weight biomass were significantly lower in W_1_ than in either drip irrigation treatment (W_2_ and W_3_), but there was no significant difference between the W_2_ and W_3_ treatments (*p* > 0.05). The results indicate that drip irrigation can significantly enhance the above- and belowground biomass of sweet potato in comparison to burrow irrigation. However, there was no discernible impact of irrigation volume on biomass under the drip irrigation method (Table 1).

#### 2.2.3. LAI

LAI increased linearly with seedling fertility in all treatments, with a gradual increase in the difference between the W_2_ and W_3_ treatments. The LAI was highest in the W_3_ treatment at all fertility stages, generally decreasing in the order W_3_ > W_2_ > W_1_. At 27 days; the LAI was significantly higher under drip irrigation than under hole irrigation (*p* < 0.05); whereas there was no significant difference between W_2_ and W_3_ (*p* > 0.05). At 50 days, the LAI was significantly higher in W_3_ than in W_1_ and W_2_ (*p* < 0.05) but with no significant difference between optimized drip irrigation and hole irrigation (*p* > 0.05; Table 2).

#### 2.2.4. Photosynthetic Parameters

At 27 days after transplantation, the seedling net photosynthetic rate (Pn), intercellular CO_2_ content (C_i_), and transpiration rate (Tr) varied significantly among treatments (*p* < 0.05). However, as the fertility period progressed, the W_1_ treatment exhibited significantly lower values than both the W_2_ and W_3_ treatments (*p* < 0.05). Differences between the W_2_ and W_3_ treatments gradually decreased throughout the study period until there was no significant difference (*p* > 0.05). There were no significant differences in stomatal conductance (Gs) between the W_2_ and W_3_ treatments (*p* > 0.05), and the differences gradually decreased between the drip irrigation and hole irrigation treatments. At all fertility stages, water utilization efficiency (WUE) was significantly higher in W_1_ than in the other treatments (*p* < 0.05), with no significant difference between W_2_ and W_3_ (*p* > 0.05; Table 3).

### 2.3. Treatment Effects on Seedling Roots Morphology

At 14 days after transplantation, root morphology indices were significantly higher under hole irrigation than under drip irrigation (*p* < 0.05), with no significant difference between the optimized and conventional drip irrigation treatments (*p* > 0.05). The total root length, root surface area, and root volume of sweet potato seedlings increased with the amount of irrigation water supplied, and the highest number of root tips was observed in the W_2_ treatment. At 27 days, root morphology indices were higher in the W_2_ treatment than in the other treatments, and total root length and root volume differed significantly from those measured at 14 days (*p* < 0.05). At 39 days, all treatments exhibited a trend of increasing and then decreasing with increasing irrigation. Root surface area, root volume, and root tip number differed significantly among treatments (*p* < 0.05). Total root length was significantly lower at 39 days than at 14 days, and the W_1_ and W_2_ treatments differed significantly between both time points (*p* < 0.05; Figure 3).

### 2.4. Treatment Effects on Seedling ^13^C Accumulation and Partition Rates

At all sampling points, the ^13^C accumulation and partitioning rates, numbers of tuberous roots, total ^13^C accumulation, and root–crown ^13^C accumulation ratio were significantly higher in the W_2_ treatment than in the other treatments. Among seedling organs, ^13^C accumulation was lowest in fibrous roots, and among treatments, it was highest in W_2_. Differences between treatments gradually decreased with plant growth, eventually becoming non-significant (*p* > 0.05). ^13^C accumulation in tubers and aboveground parts was significantly higher in W_2_ than in the other treatments. ^13^C accumulation rates were significantly higher in W_1_ than in the other treatments (*p* < 0.05), whereas there was no significant difference between treatments W_2_ and W_3_ (*p* > 0.05). The root–crown ^13^C accumulation ratio was significantly higher under optimized drip irrigation than under either hole irrigation or traditional drip irrigation (*p* < 0.05). Total ^13^C accumulation was highest in W_2_, followed by W_3_ and W_1_, with significant differences among treatments (*p* < 0.05; Table 4). These results indicated that moderate irrigation facilitated the transfer of photosynthetic products into sweet potato tubers. However, both excessive and insufficient irrigation impeded ^13^C accumulation and distribution.

### 2.5. Treatment Effects on Sweet Potato Yield Components and Deformation

#### 2.5.1. Sweet Potato Yield

Sweet potato final yield and yield component indices were significantly higher under optimized drip irrigation than either conventional drip irrigation or hole irrigation. Optimized drip irrigation increased the yield by 42.98% and 4.76% compared with hole irrigation and traditional drip irrigation, respectively, although the difference between the drip irrigation treatments was not significant. Sweet potato weight per plant and potato yield per plant differed significantly between hole irrigation and optimized drip irrigation (*p* < 0.05), with no significant difference between hole irrigation and conventional drip irrigation (*p* > 0.05; Table 5).

#### 2.5.2. Deformation

Sweet potato deformation and specific gravity were significantly higher in W_2_ than in W_1_ and W_3_ (*p* < 0.05), with no significant difference between W_1_ and W_3_ (*p* > 0.05). However, in both drip irrigation treatments, the deformity rate increased significantly as the irrigation amount increased (*p* < 0.05) until that under optimized drip irrigation was significantly lower than that under the other treatments (*p* < 0.05), with no significant difference between hole irrigation and traditional drip irrigation (*p* > 0.05; Table 6). This result indicated that over-irrigation adversely affected sweet potato morphology under both subsurface and drip irrigation.

### 2.6. Redundancy Analysis (RDA) of Soil Environment, Root Morphology, and Various Physiological Indicators in the Seedling Root Zone

RDA was conducted to examine correlations between root zone soil environment indicators and root morphology indicators among treatments. The results showed that the root zone soil environment and root morphology were significantly positively correlated with the drip irrigation treatment and negatively correlated with the hole irrigation treatment (Figure 4). These results indicated that the irrigation method was an important factor influencing the soil environment and seedling root growth.

Correlation analysis also showed that the seedling leaf Pn, Ci, Gs, above- and belowground dry weight, and sweet potato yield were significantly positively correlated with the W_2_ treatment (*p* < 0.05), whereas the deformity rate, LAI, Tr, and WUE were significantly positively correlated with the W_3_ treatment (*p* < 0.05). All indicators were negatively correlated with the W_1_ treatment, except for the deformity rate. The deformity rate was significantly positively correlated with the W_1_ treatment (*p* < 0.05) and negatively correlated with all other indicators (Figure 5). Thus, irrigation treatment was an important factor influencing photosynthesis, carbon transfer, and yield in sweet potato seedlings.

## 3. Discussion

### 3.1. Treatment Effects on Soil Moisture-Oxygen-Temperature in the Root Zone

This is the first study to monitor the VWC, temperature, CO_2_ content, and root morphology of sweet potato seedlings in the field during the post-transplantation rooting period. Penman’s formula is employed to calculate the quantity of irrigation required based on the actual crop evapotranspiration (ETc). This necessitates the consideration of meteorological factors, including light, temperature, and water vapor saturation. However, in this experiment, only a single irrigation was conducted, and ETc was not employed to calculate the irrigation volume. The results showed that the soil moisture content fluctuated, with a decreasing trend throughout the reproductive period, except during extreme weather events that caused significant increases or decreases in moisture content. Soil VWC did not differ significantly among the two drip irrigation treatments in the first month; however, all drip irrigation treatments had significantly higher water contents than hole irrigation. From 30 to 50 days after transplantation, VWC differed significantly among treatments and increased continuously with the amount of water supplied. This finding is consistent with those of previous studies [16,17,18] and suggests that increasing irrigation volume leads to higher soil VWC. However, in both drip irrigation treatments, VWC did not differ significantly between 0 and 30 days after transplantation. This phenomenon is attributable to the sandy loam soil at the study site. Single-application hole irrigation provides a large volume of water within a short time span, which can lead to rapid water infiltration and loss. In contrast, drip irrigation allows water to slowly penetrate the soil, maintaining the VWC within a certain range [19,20].

Good soil aeration is crucial for root differentiation and growth. Roots differentiate easily in well-aerated soil; whereas in poorly aerated soils, excessive CO_2_ can be toxic to the root system, hindering growth [21]. Our results indicate that the soil CO_2_ content was significantly higher in the drip irrigation treatments than under hole irrigation within the first month after transplanting. However, differences between treatments were significant at 30–50 days after transplanting, and CO_2_ concentrations were significantly higher in W_3_ than in W_2_ and W_1_. An increase in drip irrigation volume can hinder soil aeration, leading to a reduction in O_2_ content and an increase in CO_2_ content, which can have an adverse effect on root system growth, ultimately impacting tuber yield [22]. A previous study demonstrated that proper aeration promotes the transport of photosynthesis assimilation products from the leaves to tuberous roots, increasing tuber yield [23]. The amount of irrigation water supplied directly affects soil aeration and indirectly affects the degree of soil compactness, which, in turn, affects the expansion of the sweet potato root system. A larger irrigation volume leads to a greater soil firmness, which can cause sweet potato deformation [24,25]. In this study, the length and volume of sweet potato roots were significantly lower under hole irrigation than under drip irrigation because hole irrigation is performed in a single application, which causes a sudden increase in the soil VWC in the root zone, increasing the localized mechanical resistance that inhibits sweet potato root growth [26].

Sweet potatoes are a thermophilic crop, such that temperature plays a crucial role in their growth. We detected no significant difference in soil temperature among treatments, as soil temperature is heavily influenced by air temperature, and at a certain air temperature, soil moisture affects soil temperature [27]. The main reason for the effect of soil moisture on soil temperature is that the specific heat capacity of water is greater than that of soil; as the irrigation volume increases, so do the soil water content and specific heat capacity of the soil, resulting in a gradual decrease in soil temperature [28]. Li Siping analyzed the relevant indicators of sweet potato in the middle growth stage through RDA, indicating that RDA can characterize the relationship between soil water, air heat, and treatment [29]. In this experiment, a single irrigation event was employed, and meteorological conditions did not influence soil water content or CO₂ concentration. However, soil temperature was found to be more susceptible to climatic influences [30]. Overall, drip irrigation maintains the granular structure of the soil and contributes to preserving the quality of water, fertilizer, gases, and heat within the soil. Drip irrigation is also the most effective method for achieving soil water and gas phase coordination [31]. Because all soil factors are interdependent, it is necessary to consider the comprehensive influence of all factors on sweet potato growth and yield to select a suitable irrigation method that will create a favorable growth environment, which will ultimately result in high tuber yield.

### 3.2. Treatment Effects on Sweet Potato Seedling Growth

Insufficient soil moisture can impede the elongation and expansion of the sweet potato root system, which can affect root differentiation. Conversely, excessive soil moisture will not further increase the values of root system morphological indicators and can lead to root rot and seedling death [32]. Our results showed that sweet potato root system morphological indices were highest in the W_2_ treatment at 50 days. Increases in LAI and water supply directly affect plant photosynthesis rates [33]. We found that LAI did not significantly differ between the W_2_ and W_3_ treatments from 0 to 10 days after transplanting. However, after 20 days, the W_2_ treatment exhibited a rapid growth trend, with a significantly higher growth rate than the W_3_ treatment. The amount of irrigation water had a significant impact on photosynthesis in the sweet potato seedlings, with significantly higher Pn and Ci values in the W_3_ treatment, which received a larger amount of drip irrigation than the other treatments. A previous study also reported that over-irrigation led to a linearly increasing (decreasing) trend in the transfer and distribution of aboveground (belowground) sweet potato photosynthesis products [34]. Another study reported that WUE peaked at 50% of the field water-holding capacity and decreased as the soil water content increased, with no significant difference at 75% and 100% of the field water-holding capacity, suggesting that WUE does not continue to increase with soil water content [35]. Similarly, we observed the highest WUE in the W_1_ treatment and no significant difference between W_2_ and W_3_ (*p* > 0.05).

Increasing the amount of water used for seedling irrigation led to an increase in both above- and belowground seedling dry weight at 50 days after transplanting. However, the highest final yield at 150 days was observed under the W_2_ treatment. This result was attributed to the susceptibility of aboveground sweet potato stems and leaves to wilting under over-irrigation, which decreases tuber yield [36]. At 50 and 150 days after transplanting, ^13^C accumulation and partitioning rates were significantly higher in belowground seedling parts under the W_2_ treatment than in the other treatments. At 50 days, aboveground plant parts showed fluctuating growth rates during overflooding, as confirmed by the ^13^C accumulation and partitioning rates at 150 days after transplanting. Aboveground growth was significantly greater in the W_3_ treatment than in the other treatments. In contrast, the under-irrigated treatment restricted sweet potato growth, negatively impacting yield. A previous study reported that sweet potato yield was significantly higher in well-aerated soils and could be significantly improved in poorly aerated soils through ^13^C-labeling of assimilates [37,38]. Thus, moderate irrigation facilitates the establishment of the sweet potato root system and improves nutrient transport from aboveground to belowground plant parts.

Excessive drip irrigation can adversely affect plant morphology and crop yield. Determining the optimal drip irrigation supply supports high plant quality and yields, while conserving water resources [39]. Although we detected no significant difference in seedling survival rates between traditional and optimized drip irrigation treatments in this study, the final yield and the rate of commercially marketable sweet potatoes were significantly higher in the W_2_ treatment than in the W_3_ treatment, with a significantly higher rate of tuber deformation in the W_3_ treatment. As we did not monitor indices for mid- and late-stage sweet potato growth in this study, future research should explore the effects of different irrigation quantities on sweet potato agronomy and quality from molecular and genetic perspectives.

## 4. Materials and Methods

### 4.1. Experimental Samples and Sample Plots

The experiment was planted on 18 May 2021 and harvested on 18 October 2021. The main fresh-eating sweet potato variety in northern China, Yanshu 25, was selected. A field experiment was conducted at the Nancun Experimental Base of Qingdao Agricultural University (36°26′ N, 120°04′ E). The area has a warm, temperate, semi-humid monsoon climate, with precipitation concentrated in July to August, rain and heat at the same time, an average annual temperature of 12–14 °C, and an average annual rainfall of 630.98 mm. The soil type of the experimental site is brown loam, and the basic physical and chemical properties of the soil are shown in Table 7. The meteorological station was located 1 km from the experimental site. Average monthly rainfall and temperature in Namchon Town for the past three years is shown in Table 8.

### 4.2. Experimental Design

Three treatments were set up in the experiment: hole irrigation (i.e., “watering hole water”). The amount of water for the local farmers was about 800 mL per plant, 5 × 10^4^ seedlings per hectare, and the total amount of water for the fixed seedlings was calculated to be about 39.6 m^3^ ha^−1^, which was recorded as W_1_. Optimized drip irrigation treatment was recorded as W2, with a volume of 180 m^3^ ha^−1^. This was calculated according to the water holding capacity of the field, which was 65–70%. Traditional drip irrigation treatment was recorded as W3, with a volume of 450 m^3^ ha^−1^. This was calculated according to the water holding capacity of the field, which was 85–90%. The three treatments represent the past, future, and present water management methods of sweet potato after transplanting. A randomized block design, each treatment set up three replicates, and a total of nine plots were set up, with a single plot area of 30 m^2^ (6 m × 5 m), each plot having 5 ridges, plant spacing of 0.2 m, and ridge spacing of 0.85 m. Before planting, nitrogen fertilizer (CO(NH_2_)_2_), phosphate fertilizer (K_2_HPO_4_), and potassium fertilizer (K_2_SO_4_) were applied, and the dosages were 120 kg hm^−2^, 75 kg hm^−2^, and 150 kg hm^−2^, respectively, as base fertilizer.

### 4.3. Measuring Indicators and Methods

#### 4.3.1. Dynamic Monitoring of Soil Volumetric Water Content, Soil Temperature, and Soil CO_2_ Concentration

We used soil temperature and humidity sensors (HJH-WSD-02, Weihai, China) to adopt a stratified monitoring method, setting a soil moisture and temperature observation point every 15 cm from the top of the ridge to dynamically monitor soil moisture and temperature within the corresponding range. Prior to transplanting sweet potatoes, the sensor probe was inserted at the appropriate position on the ridge of each plot according to the set observation points, and volumetric soil moisture and soil temperature were measured at 0–30 cm. The soil temperature and soil moisture sensors were set to collect data every 30 min. We used a soil CO_2_ sensor (HJH-CO_2_-01, Weihai, China) to monitor soil CO_2_ levels in a stratified manner, with a soil CO_2_ monitoring point set every 15 cm from the top of the ridge to dynamically monitor soil CO_2_ levels in the corresponding area. The device was inserted between the plants on the ridge of each plot prior to sweet potato transplanting to measure soil CO_2_ at 0–30 cm. The sensor was set to collect data every 30 min. All sensors were connected to the data collection internet of things (IOT) (HJH-WG-01, Weihai, China) gateway and set to upload data every 30 min.

#### 4.3.2. Determination of Physiological Indices and Photosynthetic Parameters of Sweet Potato

Sweet potato biomass determination: Five sweet potato plants were randomly selected from each sample plot at 27, 39, and 50 days after transplanting. The fresh weight was obtained by separating the fresh stems and leaves from the stored roots, and then drying them in an oven at 105 °C for 30 min and at 80 °C for 48 h until a constant weight was achieved. The dry weight was then determined [40,41].

Leaf area index (LAI): For each treatment, we selected uniform and representative plants, removed all leaves, and weighed the petioles. The perforation method was used to determine [42].

Determination of photosynthetic parameters of sweet potato: at 9:00–11:00 in the morning, using CIRAS 3 type photosynthetic apparatus, each treatment measured 5 strains of sweet potato main stem at the top of the 4th to 5th fully expanded leaves. Net photosynthetic rate (Pn), transpiration rate (Tr), stomatal conductance (Gs), and intercellular CO_2_ concentration (Ci) index were measured [43].

The ^13^C labeling method of sweet potato leaves and the determination of ^13^C accumulation and distribution rate: The labeling test was conducted on sweet potato leaves 42 days after transplanting. Ba^13^CO_3_ reagent was reacted with phosphoric acid in a reaction device to produce ^13^CO_2_. The labeled ^13^CO_2_ gas was then collected in a bag. Representative plants with uniform growth were selected from each plot. The ^13^CO_2_ was labeled on the 4th to 5th fully expand leaves of the longest main stem of the plants between 9:00 and 11:00 on a sunny day. The measurement was taken using the gas bag method [44,45]. The δ^13^C and total carbon content (C %) were determined via IRMS in the stable isotope laboratory of the Department of Plant Science, University of California, Davis, CA, USA.

#### 4.3.3. Determination of Root Morphological Indices of Sweet Potato

Samples were collected on the 14th, 27th, 39th, and 50th days after transplanting. Immediately after field sampling, sweet potato root systems were washed with distilled water. All root systems were scanned using an Epson v850 Pro root scanner(v850 Pro, Epson Inc., Nagano Prefecture, Japan) with a resolution of 400 bpi. Root morphological parameters were analyzed using Win RHIZO software (WinRHIZO 2024a) [46].

#### 4.3.4. Determination of Survival Rate of Potato Seedlings, Yield and Deformation

Seedling survival rate: The number of surviving potato seedlings in each plot was counted, and the ratio of surviving seedlings to the total number of transplants one week after transplantation was taken as the survival rate of potato seedlings.

Sweet potato commercialization rates: Sweet potato are classified into three grades based on the weight of their storage roots: Grade A, weighing 150–600 g; Grade B, weighing more than 600 g; and Grade C, weighing less than 150 g (Big Data Platform for Sweet Potato Industry; http://www.isweetpoto.jaas.ac.cn, 2 April 2024). Marketable sweet potato storage roots are classified into grades A and B, with Grade A having the highest economic value and being of the highest quality. Grade C storage roots are generally considered to have low market value. The commercial potato rate is determined by calculating the percentage of Grade A and Grade B storage roots.

Deformity rate: 150 plants were randomly selected from each treatment at the time of harvest to count the proportion of pieces with deformities (split, irregular shape, etc.), and the deformity rate of sweet potato was obtained statistically.

Final yield: At the time of harvest, the number of individual plants and the fresh weight of tubers were recorded for each sample in each treatment. The total weight was then weighed in order to calculate the yield.

#### 4.3.5. Calculation Formula

Seedling survival rate = number of seedlings surviving after transplanting/total number of transplants

Sweet potato commercialization rate = number of commercial potatoes/total number of sweet potato

Sweet potato deformity rate = number of deformed potatoes/total number of sweet potato

^13^C isotope ratio within the sample R_sample_ = (δ^13^C/1000 + 1) × R_scale_ (R_scale_ is the carbon isotope ratio of the standard material, R_scale_ = 1.0783)

^13^C accumulation in each organ = R_sample_/(R_sample_ + 1) × C% × biomass dry weight (g)

^13^C allocation rate (%) = ^13^C accumulation in the organ/total ^13^C accumulation in the plant × 100%

### 4.4. Data Analysis

The data were organized using Excel 2019 software. To clarify the differences between treatments, mathematical and statistical analyses of the various indicators and related data were performed using SPSS 26.0. The LSR method was used to compare the significance of the differences between the means of the indicators of each treatment. To clarify the correlation between treatments on soil water, air, and heat in the root zone and sweet potato growth, we performed RDA (redundancy) analysis using Canoco 5.0.

## 5. Conclusions

The results of the experiment confirmed the hypothesis that soil moisture directly affects sweet potato growth and yield, while temperature plays an indirect role. Compared to hole irrigation, drip irrigation improved the root zone soil hydrothermal and root morphology properties, LAI, and photosynthetic capacity of sweet potato seedlings in this study. These improvements enhanced sweet potato seedling growth, as well as tuber quality and yield.

Moderate drip irrigation effectively decreased the rate of sweet potato deformation, promoting commercial yield and the transport of photosynthetic products from aboveground to belowground plant parts. However, excessive drip irrigation is not conducive to sweet potato growth but rather causes an imbalance of reservoir sources and wastes water by promoting vigorous aboveground growth. Therefore, we recommend a spring drip irrigation rate of 180 m^3^ ha^−1^ for sweet potato cultivation in northern China.

## Figures and Tables

**Figure 1 plants-13-01561-f001:**
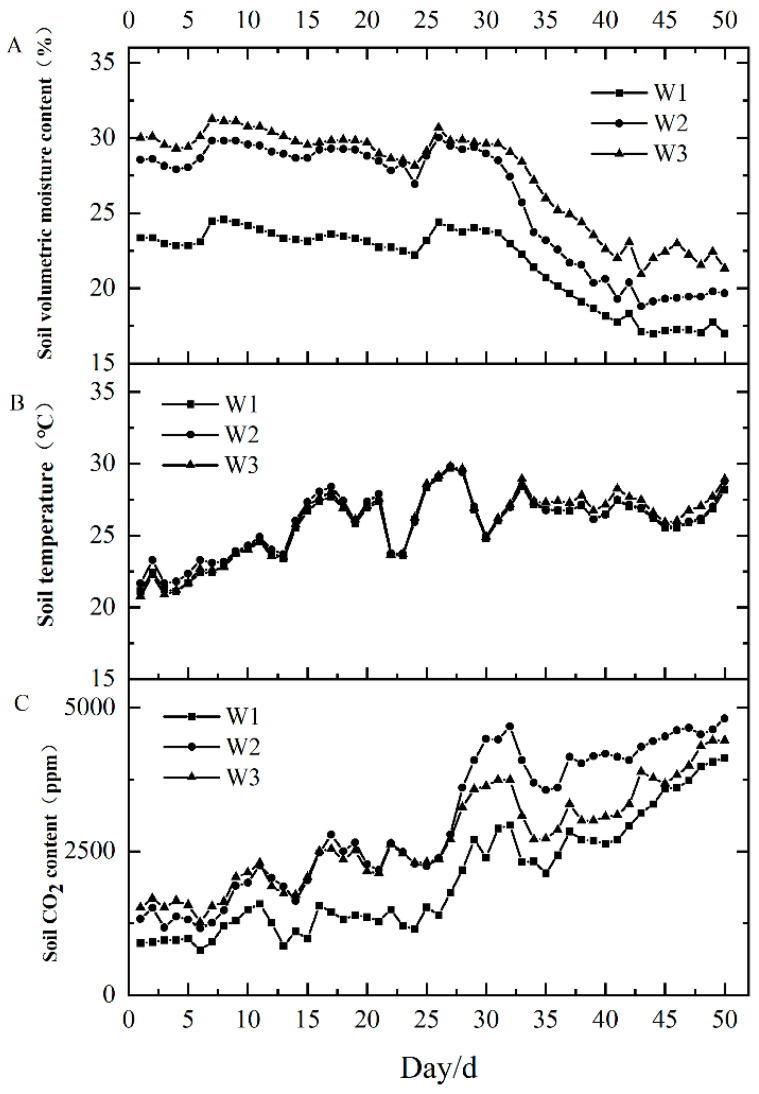
Dynamic variation in the soil volume moisture content, temperature, and CO_2_ content at 0–50 days. Note: (**A**) soil volume moisture content (%); (**B**) soil temperature (°C); (**C**) soil CO_2_ content (ppm). W_1_, W_2_, and W_3_, respectively, represent different experimental treatments. In this study, the effect of soil water vapor heat on stratification at 15 and 30 cm depths was investigated. However, it was found that the change rule of each factor at 15 and 30 cm depths was identical; thus, the use of an average value for soil water vapor heat to express the change rule.

**Figure 2 plants-13-01561-f002:**
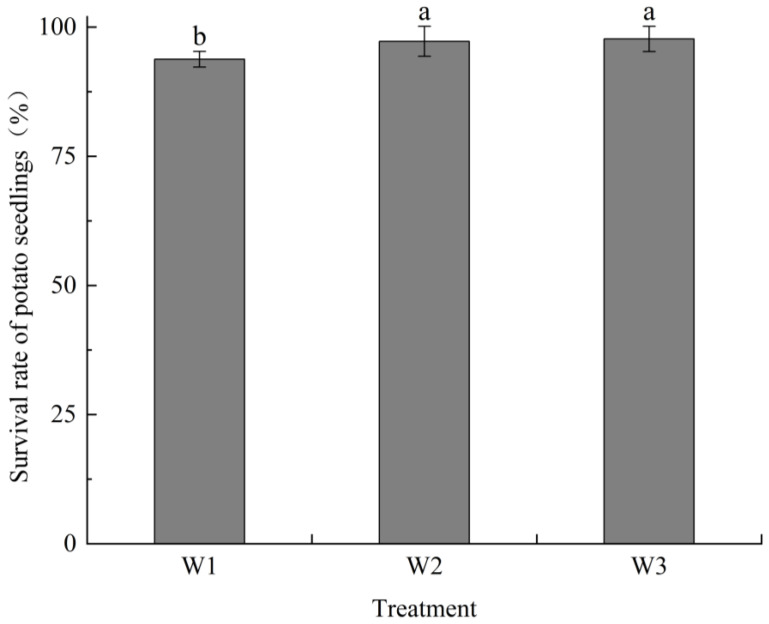
Survival rate of sweet potato seedlings after transplanting under different treatments. Different lowercase letters indicate significant differences (*p* < 0.05).

**Figure 3 plants-13-01561-f003:**
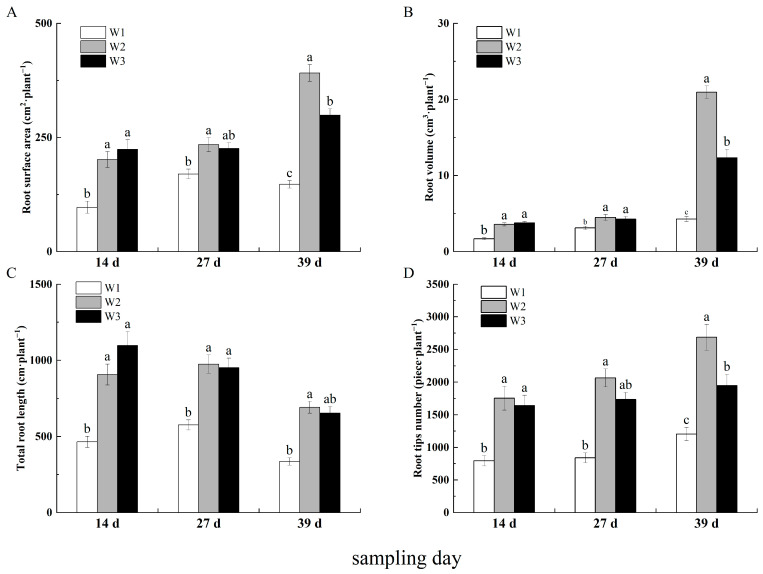
Variation in the root morphological parameters under different treatments. Note: (**A**) root surface area (cm^2^ plant^−1^); (**B**) root volume (cm^3^ plant^−1^); (**C**) total root length (cm plant^−1^); (**D**) toot tips number (piece plant^−1^). W_1_, W_2_, and W_3_ represent different treatments. Different lowercase letters in each treatment showed significant difference (*p* < 0.05).

**Figure 4 plants-13-01561-f004:**
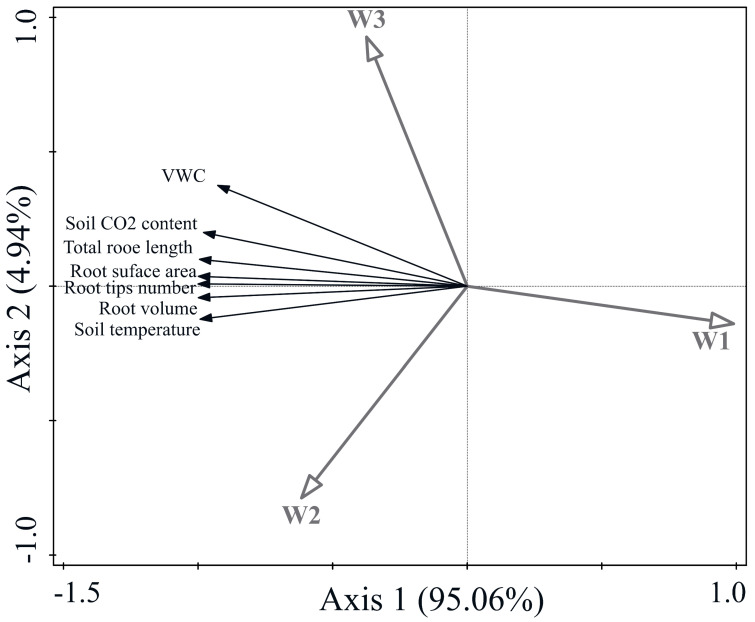
RDA analysis of soil internal environment and root morphological indices under different treatments. The coordinates of the first axis in the figure explain 95.06% of the variance, respectively, and the significances (according to Monte Carlo permutation tests) of all canonical axes were PA = 0.012, PB = 0.018, and PC = 0.026.

**Figure 5 plants-13-01561-f005:**
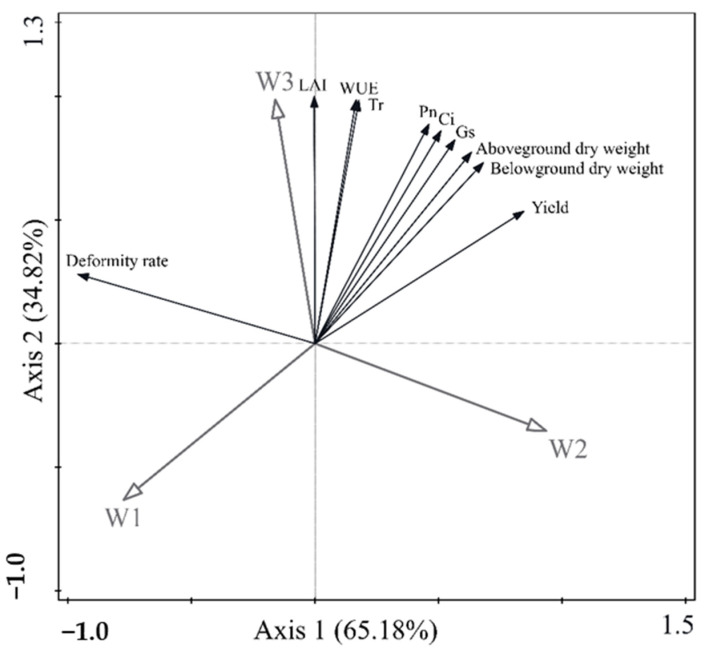
RDA analysis of physiological indices and agronomic indices of sweet potato under different treatments. The coordinates of the first axis in the figure explain 65.18% of the variance, respectively, and the significances (according to Monte Carlo permutation tests) of all canonical axes were PA = 0.012, PB = 0.018, and PC = 0.026.

**Table 1 plants-13-01561-t001:** Changes in aboveground and belowground biomass under different treatments.

Part	Treatment	Sampling Day		
		27 d	39 d	50 d
Aboveground dry weight (g plant^−1^)	W_1_	6.22 ± 0.58 b	15.82 ± 1.85 b	45.11 ± 3.82 b
	W_2_	11.33 ± 1.60 a	23.57 ± 1.64 a	58.09 ± 3.48 a
	W_3_	11.58 ± 1.25 a	26.53 ± 2.51 a	62.12 ± 4.61 a
Belowground dry weight (g plant^−1^)	W_1_	0.31 ± 0.02 b	2.39 ± 0.22 b	13.55 ± 1.64 b
	W_2_	0.40 ± 0.03 ab	9.12 ± 0.67 a	22.53 ± 2.05 a
	W_3_	0.54 ± 0.15 a	10.06 ± 0.85 a	24.56 ± 2.20 a

Note: The data in the table are the mean ± standard deviation. Different lowercase letters in the same column indicate significant differences (*p* < 0.05).

**Table 2 plants-13-01561-t002:** Variations in the leaf area index under different treatments.

Treatment	Sampling Day		
	27 d	39 d	50 d
W_1_	0.73 ± 0.18 b	1.27 ± 0.29 b	2.41 ± 0.25 b
W_2_	1.08 ± 0.22 a	1.53 ± 0.23 ab	2.70 ± 0.20 b
W_3_	1.40 ± 0.21 a	2.06 ± 0.47 a	4.53 ± 0.76 a

Note: The data in the table are the mean ± standard deviation. Different lowercase letters in the same column indicate significant differences (*p* < 0.05).

**Table 3 plants-13-01561-t003:** Variations in the photosynthetic parameters under different treatments.

Sampling Day	Treatment	Pn (µmol·m^−2^·s^−1^)	Ci (µmol·mol^−1^)	Gs (mmol·m^−2^·s^−1^)	Tr (µmol·m^−2^·s^−1^)	WUE (µmol·mmol^−1^)
27 d	W_1_	11.14 ± 1.08 c	59.36 ± 4.25 c	143 ± 12.54 b	2.06 ± 0.14 c	4.95 ± 0.19 a
	W_2_	13.48 ± 1.13 b	76.84 ± 5.18 b	162.85 ± 11.54 a	2.15 ± 0.18 b	4.15 ± 0.27 b
	W_3_	14.26 ± 1.24 a	84.26 ± 6.24 a	176.48 ± 12.42 a	2.34 ± 0.12 a	3.86 ± 0.24 b
39 d	W_1_	13.78 ± 1.05 b	72.57 ± 11.43 c	150.50 ± 19.90 b	2.45 ± 0.17 b	5.15 ± 0.29 a
	W_2_	14.96 ± 1.44 ab	92.80 ± 3.29 b	172.29 ± 11.96 a	2.76 ± 0.16 a	4.96 ± 0.38 b
	W_3_	15.45 ± 1.90 a	117.00 ± 5.82 a	184.23 ± 13.62 a	2.98 ± 0.12 a	4.42 ± 0.32 b
50 d	W_1_	13.00 ± 1.16 b	138.60 ± 12.78 b	170.64 ± 15.99 a	3.70 ± 0.46 b	5.40 ± 0.47 a
	W_2_	15.22 ± 1.29 a	154.43 ± 11.09 a	187.52 ± 10.85 a	3.78 ± 0.19 a	5.03 ± 0.48 b
	W_3_	16.88 ± 1.35 a	163.64 ± 9.67 a	194.83 ± 9.90 a	3.94 ± 0.23 a	4.85 ± 0.51 b

Note: The data in the table are the mean ± standard deviation. Different lowercase letters in the same column indicate significant differences (*p* < 0.05).

**Table 4 plants-13-01561-t004:** Effects of different treatments on the ^13^C accumulation and distribution rate of sweet potato.

Sampling Day	Treatment	Accumulation µg·Plant^−1^ (Allocation %)			Accumulation Root–Shoot Ratio R/S	Total Accumulation (µg·plant^−1^)
		Fibrous Root	Storage Root	Aboveground Tissues		
50 d	W_1_	1.54 ± 0.12 a (10.13)	3.26 ± 0.61 c (21.44)	10.40 ± 1.18 b (68.43)	0.46	15.20 ± 1.91 c
	W_2_	1.13 ± 0.19 b (4.61)	8.97 ± 0.93 a (36.00)	14.41 ± 0.52 a (59.39)	0.70	24.51 ± 1.64 a
	W_3_	1.42 ± 0.20 ab (6.97)	5.29 ± 0.49 b (25.98)	13.65 ± 1.23 a (67.05)	0.49	20.36 ± 1.28 b
150 d	W_1_	0.22 ± 0.03 a (1.02)	14.53 ± 1.65 c (67.52)	6.77 ± 1.24 b (37.46)	2.17	21.52 ± 1.53 c
	W_2_	0.17 ± 0.02 a (0.47)	26.08 ± 2.25 a (72.71)	9.62 ± 0.56 a (26.82)	2.72	35.87 ± 2.65 a
	W_3_	0.19 ± 0.02 a (0.65)	19.73 ± 1.83 b (67.25)	9.42 ± 0.87 a (32.10)	2.11	29.34 ± 2.21 b

Note: The data in the table are the mean ± standard deviation. Different lowercase letters in the same column indicate significant differences (*p* < 0.05).

**Table 5 plants-13-01561-t005:** Differences in the yield and yield components of sweet potato under different treatments.

Treatment	Yield (t·ha^−1^)	Root Weight (g·Plant^−1^)	Number of Storage Roots Plant
W_1_	25.57 ± 2.41 b	610.28 ± 17.31 b	2.51 ± 0.30 b
W_2_	36.56 ± 2.79 a	781.68 ± 27.38 a	4.00 ± 0.12 a
W_3_	34.90 ± 5.80 a	740.40 ± 24.70 ab	2.96 ± 0.76 b

Note: The data in the table are the mean ± standard deviation. Different lowercase letters in the same column indicate significant differences (*p* < 0.05).

**Table 6 plants-13-01561-t006:** Sweet potato commodity rate, proportion of commodity potato, and deformity rate.

Treatment	Commodity Rate (%)	Proportion of Commodity Potato (%)	Deformity Rate (%)
W_1_	64.39 ± 2.01 b	68.80 ± 5.74 b	11.37 ± 3.18 a
W_2_	86.02 ± 1.59 a	85.52 ± 9.44 a	3.35 ± 2.78 b
W_3_	70.32 ±11.31 b	67.93 ± 6.07 b	10.31 ± 0.55 a

Note: The data in the table are the mean ± standard deviation. Different lowercase letters in the same column indicate significant differences (*p* < 0.05).

**Table 7 plants-13-01561-t007:** Basic physical and chemical properties of the soil.

Soil Bulk Density	Proportion of Clay Soil Sand	Electrical Conductivity	pH	Organic Matter (g·kg^−1^)	Alkaline Hydrolysis Nitrogen (mg·kg^−1^)	*p* Content (mg·kg^−1^)	K Content (mg·kg^−1^)
1.27	7:40:53	0.80	8.02	8.23	57.91	10.43	82.01

**Table 8 plants-13-01561-t008:** Average monthly rainfall and temperature in Namchon Town for the past three years.

	Jan.	Feb.	Mar.	Apr.	May.	Jun.	Jul.	Aug.	Sept.	Oct.	Nov.	Dec.
Rainfall/mm	7.64	12.33	14.56	30.67	50.7	72.68	153.26	174.2	56.58	24.95	23.99	9.41
temperatures/°C	−1.91	0.78	6.28	12.83	18.74	23.00	26.03	25.58	21.44	14.82	7.13	0.32

## Data Availability

Data derived from public domain resources.

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
