# Peer review of "Root Zone Water Management Effects on Soil Hydrothermal Properties and Sweet Potato Yield"

_plants, 2024, doi:10.3390/plants13111561_

Round 1

Reviewer 1 Report

Comments and Suggestions for Authors

The theme of the manuscript is appropriate and original. It fits into the theme of the journal Plants

The research results of the conducted experiment are very important and can have application in agricultural practice.

However, the questionable nature of the research conducted once . In agricultural practice, the results should be at least two years old, and only then the results obtained give results that can be used in practice. From the content of the studied work, the result is that this experiment was conducted only once. I would very much like to ask the authors to clarify this issue.

Abstract does not raise major objections. The contained content informs what research results were obtained. However, it is proposed to supplement it with information on at what time and where the experiment was performed.

The introduction properly presents the problematic and purpose of the research. Layout, structure and division of the content correct. The methods of research were chosen correctly they are organized in the body of the work. Manuscrypt explains how the experiment was conducted and how the material for the study was collected. The statistical methods are not objectionable and are appropriately selected. In the chapter , "Results", and in the chapter , "Discussion", the authors correctly used terminology and also skillfully and correctly carried out considerations supporting them with the research of other authors.  The work is well organized and comprehensively described. It is scientifically justified and not misleading.  The citation in the body and the Table of References need minor corrections to adjust the writing to meet the editor's requirements. Line 114 correct caption Figure 2. is ,, potato'' correct ,, sweet potato''.

The paper after corrections for clarifications and additions can be published in the journal Plants.

Author Response

Responses for comments of Reviewer #1:

  1. However, the questionable nature of the research conducted once. In agricultural practice, the results should be at least two years old, and only then the results obtained give results that can be used in practice. From the content of the studied work, the result is that this experiment was conducted only once. I would very much like to ask the authors to clarify this issue

Response:

We feel great thanks for your professional review work on our article.

We conducted the corresponding pre-test before conducting this experiment, and this research experiment is consistent with the results of the pre-test. However, the pre-experiment lacked dynamic monitoring, while this experiment added dynamic measurement of relevant indicators, and the changes in both temperature and rainfall in the past two years were consistent, which did not have a significant effect on this experiment. Therefore, only data from this experiment were used.

  1. It is proposed to supplement it with information on at what time and where the experiment was performed.

Response:

We agree with this comment.

We have revised in Page 1, Lines 9-10: add: planted on 18 May and harvested on 18 October 2021, at the Nancun Experimental Base of Qingdao Agricultural University; Page 1, Line 11: change to: Three water management treatments were tested for sweetpotato seedlings after transplanting.

  1. The citation in the body and the Table of References need minor corrections to adjust the writing to meet the editor's requirements.

Response:

We agree with this comment.

We have revised the references, as follows:

  1. Replace articles 6, 28, 30 and 31.
  2. The reference format of the reference was modified and the “MDPI” format was used.

  1. Line 114 correct caption Figure 2. is ,, potato" correct ,, sweet potato".

Response:

We agree with this comment.

We have revised in Page 4, Line 121: replace: “potato” with “sweet potato”

Reviewer 2 Report

Comments and Suggestions for Authors

1. This research is related to irrigation management techniques, which are important for this region, and contributes to this area as a case study. However, as an academic paper, it is important to describe new findings that can be generalized to crop physiology and ecology. To do so, more detailed information on the dynamics of air temperature (e.g., minimum and maximum temperatures), humidity (preferably water vapor saturation deficit), and rainfall during the test period, which have significant effects on water stress, is needed. The main reasons for this are as follows:

2. The main cause of water stress is a lack of soil moisture, but in addition to irrigation management, rainfall dynamics are also important factors that affect water stress. In addition, during periods of low rainfall, salt accumulation can also occur depending on the irrigation management method, which is a cause of water stress.

3. Strong light, high temperatures, and high water vapor saturation deficits also promote water stress caused mainly by unsuitable soil environments, so the above meteorological environment information must also be described as the experimental environment.

4. In addition, because the values ​​of soil environmental elements vary greatly depending on the soil depth, vertical distribution information for each element is required down to the depth where the roots are distributed, and for irrigation methods such as drip irrigation, numerical information on three-dimensional distribution is also needed.

5. The results of the RDA analysis shown in Figs. 4 and 5 are likely to vary significantly depending on the environmental conditions mentioned above. The novelty of the results of this study will be enhanced by comparison to similar research reports under different meteorological and environmental conditions with reference to other literature.

Author Response

Responses for comments of Reviewer #2:

  1. This research is related to irrigation management techniques, which are important for this region, and contributes to this area as a case study. However, as an academic paper, it is important to describe new findings that can be generalized to crop physiology and ecology. To do so, more detailed information on the dynamics of air temperature (e.g., minimum and maximum temperatures),humidity (preferably water vapor saturation deficit), and rainfall during the test period, which have significant effects on water stress, is needed.

Response:

We agree with this comment.

Thank you for your valuable suggestions, according to your comments we have added the relevant content and explanations.

  1. The main cause of water stress is a lack of soil moisture, but in addition to irrigation management, rainfall dynamics are also important factors that affect water stress. In addition, during periods of low rainfall, salt accumulation can also occur depending on the irrigation management method, which is a cause of water stress.

Response:

We agree with this comment.

We have revised in Page 12, Line 360: add: Table 8 Average monthly rainfall and temperature in Namchon Town for the past three years.

Jan

Feb

Mar

Apr

May

Jun

Jul

Aug

Sept

Oct

Nov

Dec

Rainfall/mm

7.64

12.33

14.56

30.67

50.7

72.68

153.26

174.2

56.58

24.95

23.99

9.41

Temperatures /℃

-1.91

0.78

6.28

12.83

18.74

23.00

26.03

25.58

21.44

14.82

7.13

0.32

  1. Strong light, high temperatures, and high water vapor saturation deficits also promote water stress caused mainly by unsuitable soil environments, so the above meteorological environment information must also be described as the experimental environment.

Response:

We agree with this comment.

Penman's formula is based on Actual crop evapotranspiration (ETc) to calculate irrigation volume, which requires meteorological factors such as light, temperatures, water vapor saturation, etc. However, this experiment was conducted under one irrigation and did not use ETc to calculate the irrigation volume. We have plans to conduct ETc based irrigation research trials.

  1. In addition, because the values of soil environmental elements vary greatly depending on the soil depth, vertical distribution information for each element is required down to the depth where the roots are distributed, and for irrigation methods such as drip irrigation, numerical information on three-dimensional distribution is also needed.

Response:

We agree with this comment.

In this experiment, we made stratified measurements of soil hydroaerobic heat, but found that the pattern of change of each factor was consistent at depths of 15cm and 30cm. At the same time, we have previously carried out the addition of data from different depths and found that this would result in an excessively long article. Therefore, the method of averaging was used to present it. Firstly, the stratified data are shown below:

Fig 1 15cm Soil Volumetric Water Content (%)

Fig 2 30cm Soil Volumetric Water Content (%)

Fig 3 15cm Soil Temperature(℃)

Fig 4 30cm Soil Temperature(℃)

  1. The results of the RDA analysis shown in Figs.4 and 5 are likely to vary significantly depending on the environmental conditions mentioned above. The novelty of the results of this study will be enhanced by comparison to similar research reports under different meteorological and environmental conditions with reference to other literature.

Response:

We agree with this comment.

In response to this question, which is similar to question 3, the current one-time irrigation was used and did not include relevant indicators such as meteorological and environmental conditions, and there are analyses related to the correlation between soil hydro-aerothermal and inter-treatment correlations in Figure 4.

Reviewer 3 Report

Comments and Suggestions for Authors

Dear Authors,

Review of the work entitled: "The influence of water management in the root zone on soil hydrothermal properties and sweet potato yield" with ID: plants-3002497. The work contains interesting results and their interpretation. However, as a reviewer, I have a few comments on the work.

1.     Abstract: The abstract lacks proper structure and utilizes inappropriate tenses. Past perfect tense should be used for describing specific research or experiments. For instance, "A series of tests were performed in the laboratory."

2.     Introduction: While the introduction provides a good overview of the topic, it should end with a clear research goal or hypothesis, including the null hypothesis, which can be tested later in the study.

3.     Methodology: The methodology should be logically consistent and precisely described, including the research tools, techniques, and procedures used, all appropriate to the study's nature.

4.     Results: The description of the results should be clear, detailed, and consistent with the collected data and analysis methods. Any discrepancies should be explained. The interpretation provided is reasonable and based on solid evidence, logical thinking, and statistical analysis.

5.     Discussion: While the discussion comprehensively addresses the impact of various factors on sweet potato growth, it overlooks other potential factors such as soil composition, salinity, pH, diseases, and pests. The short observation period, lack of molecular and genetic analysis, geographic limitations, and potential measurement errors are also notable weaknesses.

6.       Conclusions: The conclusions should be concise and generalizable.

Comments on the Quality of English Language

 Minor editing of English language required

Author Response

Responses for comments of Reviewer #3:

  1. The abstract lacks proper structure and utilizes inappropriate tenses. Past perfect tense should be used for describing specific research or experiments.

Response:

We agree with this comment.

We have revised in Page 1, Lines 13-16: change to: “The variation characteristics of soil volumetric water content, soil temperature and soil CO2 concentration in root zone were monitored in situ during 0-50 days. The agronomy, root morphology, photosynthetic parameters, 13C accumulation, yield and yield components of sweet potato were determined.”

  1. While the introduction provides a good overview of the topic, it should end with a clear research goal or hypothesis, including the null hypothesis, which can be tested later in the study.

Response:

We agree with this comment.

We have revised in Page 2, Lines 83-84: add: We predicted that water had a direct effect on the growth and yield of sweet potato, while temperature had an indirect effect.

  1. The methodology should be logically consistent and precisely described, including the research tools, techniques, and procedures used, all appropriate to the study’s nature.

Response:

We agree with this comment.

We have revised in Page 12, Lines 379-392: change to : We used soil temperature and humidity sensors (HJH-WSD-02, China) to adopt a stratified monitoring method, setting a soil moisture and temperature observation point every 15 cm from the top of the ridge to dynamically monitor soil moisture and temperature within the corresponding range. Prior to transplanting sweet potatoes, the sensor probe was inserted at the appropriate position on the ridge of each plot according to the set observation points, and volumetric soil moisture and soil temperature were measured at 0-30 cm. The soil temperature and soil moisture sensors were set to collect data every 30 minutes. We used a soil CO2 sensor (HJH-CO2-01, China) to monitor soil CO2 levels in a stratified manner, with a soil CO2 monitoring point set every 15 cm from the top of the ridge to dynamically monitor soil CO2 levels in the corresponding area. The device was inserted between the plants on the ridge of each plot prior to sweet potato transplanting to measure soil CO2 at 0-30 cm. The sensor was set to collect data every 30 minutes. All sensors were connected to the data collection internet of things (IOT) (HJH-WG-01, China) gateway and set to upload data every 30 min.; Page 13, Lines 424-426: change to : “ The number of surviving potato seedlings in each plot was counted, and the ratio of surviving seedlings to the total number of transplants one week after transplantation was taken as the survival rate of potato seedlings.”; Page 14, Lines 438-440: change to : “At the time of harvest, the number of individual plants and the fresh weight of tubers were recorded for each sample in each treatment. The total weight was then weighed in order to calculate the yield.”

  1. The description of the results should be clear, detailed, and consistent with the collected data and analysis methods. Any discrepancies should be explained. The interpretation provided is reasonable and based on solid evidence, logical thinking, and statistical analysis.

Response:

We agree with this comment.

We have revised in Page 4, Lines 123-126: add : The results indicate that drip irrigation can significantly enhance the above- and below-ground biomass of sweetpotato in comparison to burrow irrigation. However, there was no discernible impact of irrigation volume on biomass under the drip irrigation method.

  1. While the discussion comprehensively addresses the impact of various factors on sweet potato growth, it overlooks other potential factors such as soil composition, salinity, pH, diseases, and pests. The short observation period, lack of molecular and genetic analysis, geographic limitations, and potential measurement errors are also notable weaknesses.

Response:

We agree with this comment.

The composition, salinity and pH of the selected sample plots have been described in the Materials and Methods section, and the soil conditions of the sample plots are suitable for sweet potato cultivation. There were no pests or diseases present in the sample plots, which had no effect on this experiment. Plans are in place to study the effects of pests and diseases on sweetpotato physiology and yield.

  1. The conclusions should be concise and generalizable.

Response:

We agree with this comment.

We have phrased our conclusions in segments while revised in Page 14, Lines 461-462: add : The results of the experiment confirmed the hypothesis that soil moisture directly affects sweet potato growth and yield, while temperature plays an indirect role.

Round 2

Reviewer 2 Report

Comments and Suggestions for Authors

Comments on the Quality of English Language

English grammar and expressions throughout the manuscript should be checked by native-speaking researchers.
